# Understanding teamwork in rapidly deployed interprofessional teams in intensive and acute care: A systematic review of reviews

Stefan Schilling[1,2]*, Maria Armaou[1,3], Zoe Morrison[4‡], Paul Carding[5‡], Martin Bricknell[2‡], Vincent Connelly[1‡]

1 Department of Psychology, Health & Professional Development, Oxford Brookes University, Oxford, United Kingdom, 2 School of Security Studies, King's College London, London, United Kingdom, 3 School of Health Sciences, University of Nottingham, Nottingham, United Kingdom, 4 Aberdeen Business School, Robert Gordon University, Aberdeen, United Kingdom, 5 Oxford Institute of Nursing, Midwifery and Allied Health Research, Oxford Brookes University, Oxford, United Kingdom

‡ ZM, PC, MB and VC are contributed equally and substantially to this work.
* sschilling@brookes.ac.uk

**Data Availability Statement:** All relevant data are within the article and its Supporting Information files.

## Abstract

The rapid increase of acute and intensive care capacities in hospitals needed during the response to COVID-19 created an urgent demand for skilled healthcare staff across the globe. To upscale capacity, many hospitals chose to increase their teams in these departments with rapidly re-deployed inter-professional healthcare personnel, many of whom had no prior experience of working in a high-risk environment and were neither prepared nor trained for work on such wards. This systematic review of reviews examines the current evidence base for successful teamwork in rapidly deployed interprofessional teams in intensive and acute care settings, by assessing systematic reviews of empirical studies to inform future deployments and support of rapidly formed clinical teams. This study identified 18 systematic reviews for further analysis. Utilising an integrative narrative synthesis process supported by thematic coding and graphical network analysis, 13 themes were found to dominate the literature on teams and teamwork in inter-professional and inter-disciplinary teams. This approach was chosen to make the selection process more transparent and enable the thematic clusters in the reviewed papers to be presented visually and codifying four factors that structure the literature on inter-professional teams (i.e., team-internal procedures and dynamics, communicative processes, organisational and team extrinsic influences on teams, and lastly patient and staff outcomes). Practically, the findings suggest that managers and team leaders in fluid and ad-hoc inter-professional healthcare teams in an intensive care environment need to pay attention to reducing pre-existing occupational identities and power-dynamics by emphasizing skill mix, establishing combined workspaces and break areas, clarifying roles and responsibilities, facilitating formal information exchange and developing informal opportunities for communication. The results may guide the further analysis of factors that affect the performance of inter-professional teams in emergency and crisis deployment.

**Funding:** The study authors (VC, SS, ZM, PM, MB) received funding from the UKRI Economic & Social Science Research Council (ES/V015974/1) https://esrc.ukri.org/ The funders had no role in study design, data collection and analysis, decision to publish, or preparation of the manuscript.

**Competing interests:** The authors have declared that no competing interests exist.

## Introduction

During the COVID-19 pandemic, many hospitals globally had to rapidly expand acute and intensive care capacity because of an unprecedented demand for multiple care specialties. This rapid expansion of capacity on Intensive Care, Infectious Disease, and High Dependency Units, created an urgent need for skilled staff, combined with staff shortages due to infections and shielding, with nurses, physicians, allied health professionals from non-intensive, emergency, or acute care backgrounds being re-deployed [1–5]. Many COVID wards were therefore characterised by being staffed by rapidly deployed, fluid, inter-professional teams, with many staff members not previously prepared or trained for working in a high-risk environment. Recent research on COVID-19 deployed personnel has been devoted to individual health outcomes of healthcare professionals during the pandemic [6–8], and organizational interventions aimed at mitigating risks to healthcare staff [9–12]. Little research has–therefore–been devoted to the impact of COVID-19 deployment on teams' abilities to effectively work together and develop cooperative communication processes considering the increased stressors in their work environment [2, 13–15]. A large majority of research on inter-professional teams–established to provide systems of integrated care, reduce costs, or improve patient outcomes–thus far has been focused on permanent or semi-permanent teams [16–19]. According to this research, successful inter-professional or multi-disciplinary teams benefit patient satisfaction, mortality rates, length of in-patient stay, or clinical error rates [20–26]. Nevertheless, unlike during the COVID crisis, such teams usually have received some form of common training or preparation and tend to have worked together for longer periods of time.

The processes involved in forming clinical teams during crisis is an important aspect to be considered when emergency planning in healthcare, as many of the key factors of effective teamwork in such circumstances (such as common training, shared communication patterns and leadership) may not be possible during an emergency. Research in inter-professional task-organized military personnel has shown that effective teamwork is not only dependent on team structure and professional skill-mix but also on operational requirements [27–29]. Different levels of occupational background, experience, and training, exacerbated by the hazardous environment, social distancing, and personal protective equipment (PPE) may undermine communication and familiarisation with colleagues. Fluid and rapidly changing team structures may make it difficult to establish processes and care pathways, while lack of resources and equipment might increase anxiety about one's own personal safety and the ability to care for patients. While such factors may increase risk to patient outcomes, delivery of care, individual resilience, staff mental-health and retention in all staff, they may have even more enduring effects on staff least prepared or experienced in intensive care and infectious disease environments. As such it is important to assess how personnel from different backgrounds working in ad-hoc, fluid, inter-professional teams develop teamwork.

The aim of this systematic review of reviews is to examine the current evidence base on teamwork in rapidly deployed interprofessional teams in intensive and acute care settings in order to inform future deployments and support of rapidly formed clinical teams and to identify gaps in knowledge or fruitful areas of development for future research. This will be done by thematically analysing and coding systematic reviews of empirical studies. This will be supplemented by a graphical network analysis (using the *gephi* software package), allowing greater transparency in the selection of themes, visualisation of the identified themes from the reviewed papers and identification of underlying factors structuring the themes (i.e., team-internal procedures and dynamics, communicative processes, organisational and team extrinsic influences on teams, and lastly patient and staff outcomes) [30]. Following previous research on the usage of team terminology (e.g., multi-, or inter-disciplinary, multi, or inter-

professional) [31], the authors have chosen to categorise teams on the basis of their conceptual differences between the terms "profession" and "discipline" [32]. The types of teams reviewed in the subsequent synthesis are defined in Table 1.

# Method

## Search strategy

Exploratory searches of the literature were conducted using MEDLINE, Global health, CINAHL, and APA PsycINFO in January 2021, with the aim of testing different search strings, finding 179,205 results. Search strategy refinement was developed in consultation with expert research librarians and finalised upon discussion with the project team. The final search string was narrowed to *((team* or cooperation) and (health* or clinic* or acute or medic* or hospital or inter* or multi*) and review)*. Subsequently, a definitive literature search was conducted for articles published from January 2000 until February 2021, using 7 databases (MEDLINE, Global health, CINAHL, APA PsycINFO, Business Source Complete, SCOPUS and Web of Science). An overview of the search parameters is outlined in Fig 1.

## Inclusion/Exclusion criteria

The selection included only review articles of peer-reviewed empirical studies, published in peer reviewed journals, written in English, using qualitative, quantitative, or mixed methods. In addition, articles were required to meet three out of four criteria: i) focus on inter-professional/ inter-disciplinary/ multi-disciplinary teams in a healthcare setting; ii) include at least one of intensive emergency, acute and critical care settings; iii) discuss teamwork; iv) include rapidly convened teams. Studies were excluded if they did not review empirical studies, (e.g., book reviews, case reviews, clinical audits, editorials were excluded) or were singularly focused on mono-professional environments (e.g., mental health, oncology, primary, paediatric care, rheumatology, pharmacology, radiology). Similarly, reviews focusing on training or interventions were excluded if the focus of the study was on simulated or classroom-based training environments. Whilst acknowledging their utility in multi-disciplinary and inter-professional environments, reviews focusing on interventions were also excluded, due to the requirement for long-term planning involved in scenario development. Based on these inclusion and exclusion criteria, a checklist was developed to assess each article detected in the literature search (see Supporting information).

## Search process

The PRISMA framework was applied to the search process and to guide the analysis [33, 34]. Two of the authors identified 1585 titles from the initial search of seven databases that, with duplications removed, resulted in 646 articles for screening based on the above selection criteria (see Fig 1). After title screening (divided between all authors), 151 abstracts were screened

**Table 1. Definition of terms.**

| |
| --- |
| "inter-professional" and "multi-professional" are more general terms referring to practices where members of different healthcare professions (e.g., nurses, physicians, Physiotherapists) are working together to deliver care. |
| "inter-disciplinary" describes teams that consist of members from different disciplines in the same profession and whose different occupational pathways and routine work processes overlap in the process of providing care (e.g., radiologists, surgeons, physicians). |
| "mono-professionalism" or "uni-professionalism" refer to practices of teams consisting only of members of the same profession. |

**PRISMA 2020 flow diagram for new systematic reviews which included searches of databases and registers only**

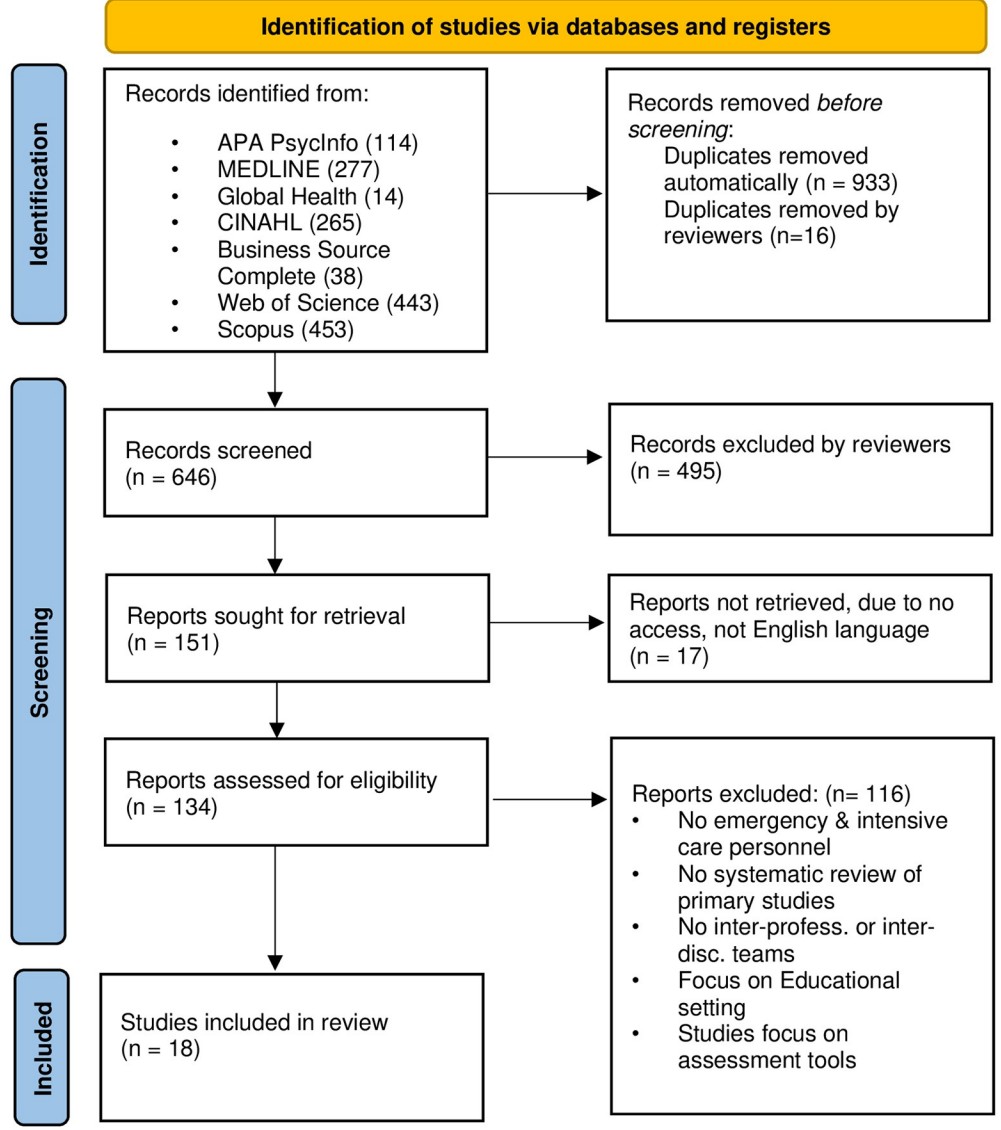

*From:* Page MJ, McKenzie JE, Bossuyt PM, Boutron I, Hoffmann TC, Mulrow CD, et al. The PRISMA 2020 statement: an updated guideline for reporting systematic reviews. BMJ 2021;372:n71. doi: 10.1136/bmj.n71

For more information, visit: http://www.prisma-statement.org/

**Fig 1. PRISMA flow chart.** The PRIMSA diagram details our search and selection process applied during the review.

for relevance. Discrepancies in results were overcome through reconciliation and discussion between the two reviewers (Author 1 and Author 2). The full texts of the remaining 134 articles were reviewed by the two reviewers and disputed articles were referred to a third reviewer to minimize bias. References of included studies were hand-searched for additional potentially relevant articles, which yielded two additional articles. During the full text review, 116 articles

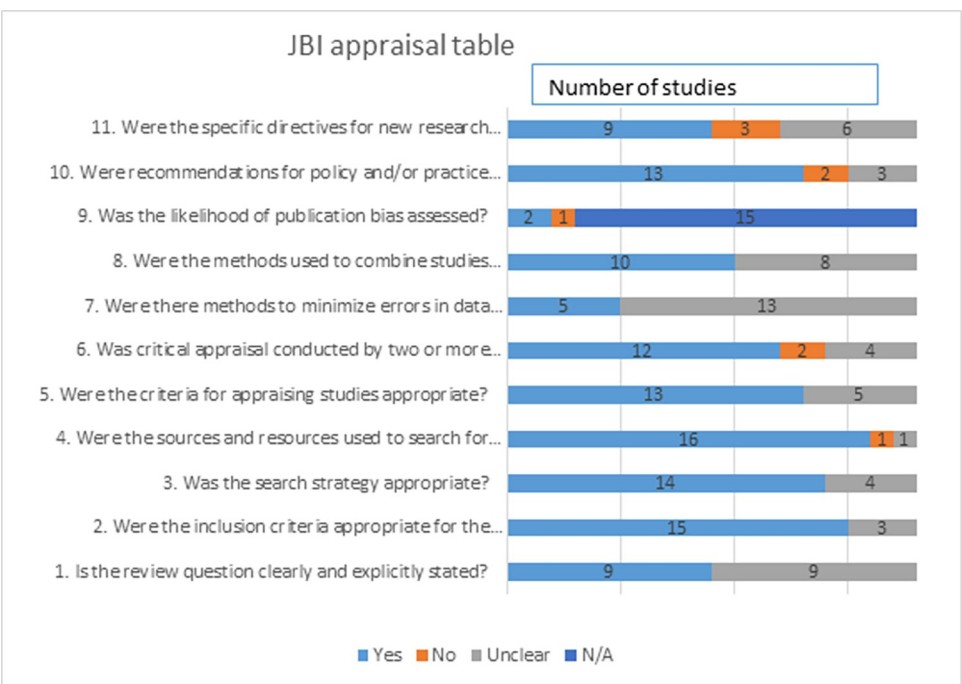

**Fig 2. Overview of JBI appraisal table.** The Figure outlines the results of using the JBI Critical Appraisal Checklist for Systematic Reviews and Research Synthesis (Copyright JBI 2020) on the 15 selected articles.

were excluded leaving 18 articles for final inclusion in our review. All identified studies were evaluated for quality using the JBI Checklist for Systematic Reviews and Research Syntheses [35, 36]. Despite differing levels of quality, none of the 18 final articles were excluded due to poor quality (See Fig 2).

## Data extraction and analysis

The process for selection is summarised in Fig 1 using the PRISMA framework [33, 34]. A checklist with the relevant study characteristics was constructed to allow a systematic data extraction in line with our research question. To this end, a data extraction tool was developed using an iterative process to document key characteristics of the 18 reviewed studies, namely:

- Research Objectives of review;

- Research Questions of review;

- Type of review (e.g., systematic, narrative);

- Databases searched by review;

- Number of included papers in review;

- Study designs of included reviews;

- Quality assessment in review;

- Key findings of review;

- Type of teams included (e.g., inter-professional, inter-disciplinary, mono-disciplinary);

- Type of occupations (e.g., nurses, physician, surgeons, allied health professionals);

- Research Settings (e.g., ICU, ED, Trauma Centre);

- Countries included in review.

Full details of the extraction table are provided in the supporting information.

The thematic data was independently extracted by two authors (Author 1 and Author 2) who are trained coders and who met to reconcile any differences through discussion. Coding was conducted thematically using NVivo 12 and NVivo for Mac (release 1.5.1) and inductive themes grouped iteratively according to the most frequent higher and lower order themes.

Contrary to other types of reviews, there are few specific reporting tools for systematic reviews of reviews [37, 38]. This systematic review of reviews was designed to include both quantitative and qualitative reviews so the process of integrative narrative synthesis, which has been frequently applied to reviews containing both quantitative and qualitative evidence [39–41] was applied to explore the relationships between the findings within and across the included studies. This process permitted an investigation of the heterogeneity of the included studies and highlighted variations that may be attributable to theoretical constructs. Specific techniques (e.g., content analysis and tabulation, concept mapping and critical reflection) were used to facilitate this knowledge synthesis process [40, 42]. The six-step iterative process is outlined in Fig 3. This permitted the organisation of reviewed studies into logical categories and the construction of a common rubric. Subsequently, concept mapping of the relationships of concepts within and between studies was used to increase methodological triangulation. The

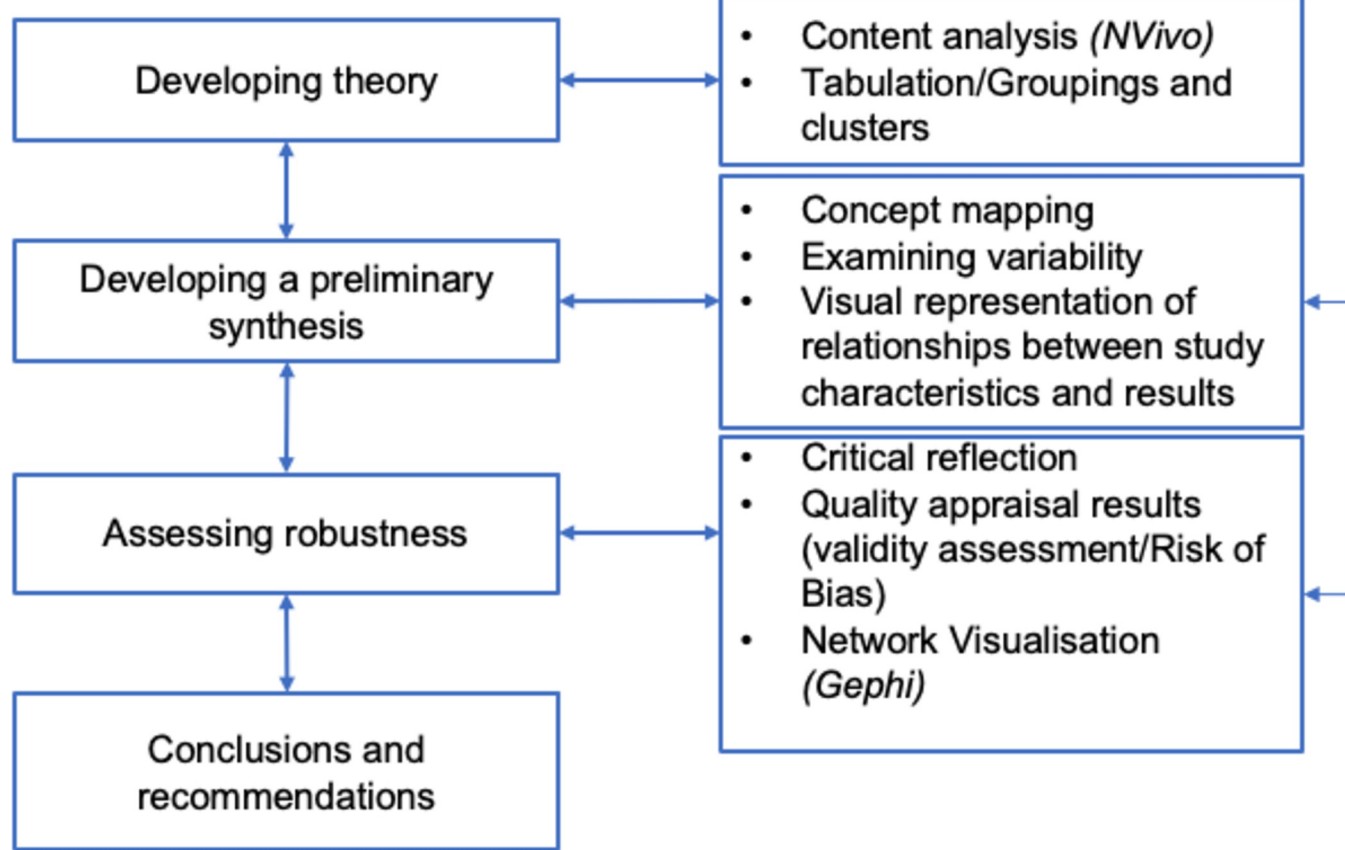

**Fig 3. Integrative narrative synthesis process.** The Figure details the integrative narrative synthesis process applied during the review (adapted from Popay et al., 2006).

subsequent codes and concepts were then presented to all the contributing authors, who discussed and spot-checked for reliability allowing for further synthesis of the themes.

To support the analysis of the data, the coded data was visualised using a graphical network analysis and exploration software programme *(Gephi release 0.92)* to graphically represent the relationships between thematic connections and identify clusters of thematic nodes, indicating overarching factors [30, 43]. The graphical network analysis was chosen, as such approaches have been found to increase transparency in qualitative data analysis, by allowing the visualization of inherent complex relationships in coded data without undermining the qualitative characteristics of the data [30, 43]. This was achieved by developing a matrix consisting of the NVivo generated themes (nodes) and the number of shared connections between themes (edges), which was formatted and uploaded into *Gephi*. The resulting network between themes was graphically visualised using the *Fruchterman Reingold* layout algorithm [44, 45] and the computation and visualization of the average weighted degree and modularity of nodes (representing clusters of nodes which themes are more closely related to each other on the basis of sharing coding references) [46]. In an iterative process, similar nodes (i.e., those sharing a substantial amount of overlap were combined to develop a new node incorporating both aspects. For example, the visualization showed that the inductive derived nodes '*occupational demands*' and '*professional roles and demands*' shared significant overlap in the graphical analysis, which led to these being merged under the new theme: '*Professional and Occupational Roles & Demands*'. Similarly, the codes '*Interpersonal relations*' and '*Interpersonal conflict*' were combined to '*Interpersonal Relations & Conflict*'. This synthesis of the themes was assessed for robustness through repeated iterative discussion with all the authors, which allowed the construction of a best evidence synthesis to inform our conclusions and recommendations.

## Results

### Review characteristics

Overall, the 18 reviews included 15 systematic reviews, with one integrative review, one scoping review and one systematised literature review. Of the 15 systematic reviews, three applied a meta-analytic approach and two utilised a meta-ethnographic approach. All reviews were published between 2006–2020 and each article reviewed between 9–98 studies (median: 26) across a variety of research designs (e.g., quasi-experimental studies, observational studies, longitudinal studies, mixed methods studies, randomized controlled trials, and qualitative studies). Fifteen of the reviews provided the geographical context of their studies and included empirical studies from the USA, Australia, the UK, Canada, New Zealand China, Israel, and numerous European countries (Italy, Belgium, Germany, Cyprus, Netherlands, Greece, Denmark, Germany, Switzerland). Only six of the 18 articles reported on individual studies from the Global South, (e.g., Egypt, South Africa, Hong Kong, Taiwan, Turkey, South Korea, Singapore, Saudi Arabia, Rwanda, Brazil, India). The review was not registered.

### Synthesis

We first tabulated the studies in terms of the type of teams that were investigated and the type of healthcare settings. The included reviews focused on aspects of healthcare teams with diverse team composition. Discrepancies in the terminology used to describe teams (e.g., multi, or inter-disciplinary, multi, or inter- professional) were frequent and add-on terms such as "*ad-hoc*" (for teams established for the purposes of the study) or "real" (for pre-existing teams) further complicated the comparisons. The clinical settings of observed teams also varied substantially. For this reason, we extracted relevant information about the terms used to describe healthcare teams, focusing on the type of healthcare professionals comprising them

**Table 2. Types of teams and setting observed in the reviewed articles.**

| Types of Teams | No of Studies |
|---|---|
| Inter-professional/ multi-professional | 13 |
| Inter-disciplinary | 6 |
| Multi-disciplinary | 5 |
| Mono/ uni-professional | **3** |
| Types of Settings | |
| Various clinical | **13** |
| Only acute/ critical care | **5** |
| Included non-healthcare setting | **1** |

and the type of settings studied (see supporting information). The included reviews most frequently examined "*interprofessional teams*", "*multi-professional teams*", "*interdisciplinary teams*", "*multidisciplinary teams*" and "*mono-professional*" or "*uni-professional*" teams (see Table 2). These reviews focused on the key characteristics of teams and in some cases highlighted differences among them but there was no unambiguous definition of terms across all the reviews. Some reviews provided valuable insights to the different facets of interprofessional, interdisciplinary, multi-disciplinary and mono-professional teamwork observed in acute, critical care or trauma settings (e.g. [47–49]). Only a small number of studies focused solely on teams within acute or critical care, while the majority reviewed evidence from a range of healthcare teams including acute or critical care teams.

## Thematic findings

The included articles assessed evidence on team characteristics and highlighted organisational and interpersonal influences on team processes. Overall, the thematic analysis synthesised the themes in the literature down to 13 themes related to teamwork and team interactions. The visual network analysis of the 13 themes showed the relative importance of each theme to the overall analysis (by visualising the number of references in the size of each node; see Fig 4. Furthermore, the modularity computation, showing the relationship between nodes, identified four distinct overarching factors represented by the four different coloured clusters of nodes (see Table 3). These factors related to 1) *Team-internal procedures and dynamics*, 2) *Communicative processes*, 3) *Organisational and team-extrinsic influences on teams*, and 4) *Team outcomes*. The network analysis shows that despite these four distinct clusters appearing, teamwork in healthcare is a messy and multi-faceted picture, where inter-personal, team-intrinsic, and team-extrinsic factors impact upon teamwork. The following section will describe the four themes identified.

## Factor 1: Team internal procedures and dynamics

The thematic analysis identified a range of themes which display in the graphical network analysis as one factor related to team internal procedures and dynamics, i.e., those factors impacted most by intra-team relationships, interactions, decisions, and processes.

**Psycho-social traits.** Despite many team-intrinsic factors being discussed in the reviewed articles as being inter-personal in nature, only five studies linked psycho-social traits and personality to teamwork [18, 49–52]. While the reviewed literature identified a range of individual characteristics impacting on teamwork (e.g., emotional intelligence, confidence, knowledge, fatigue, rudeness [50, 51, 53]), such individual characteristics cannot be fully differentiated from intra-team dynamics and structural and organisational aspects of teamwork. As such,

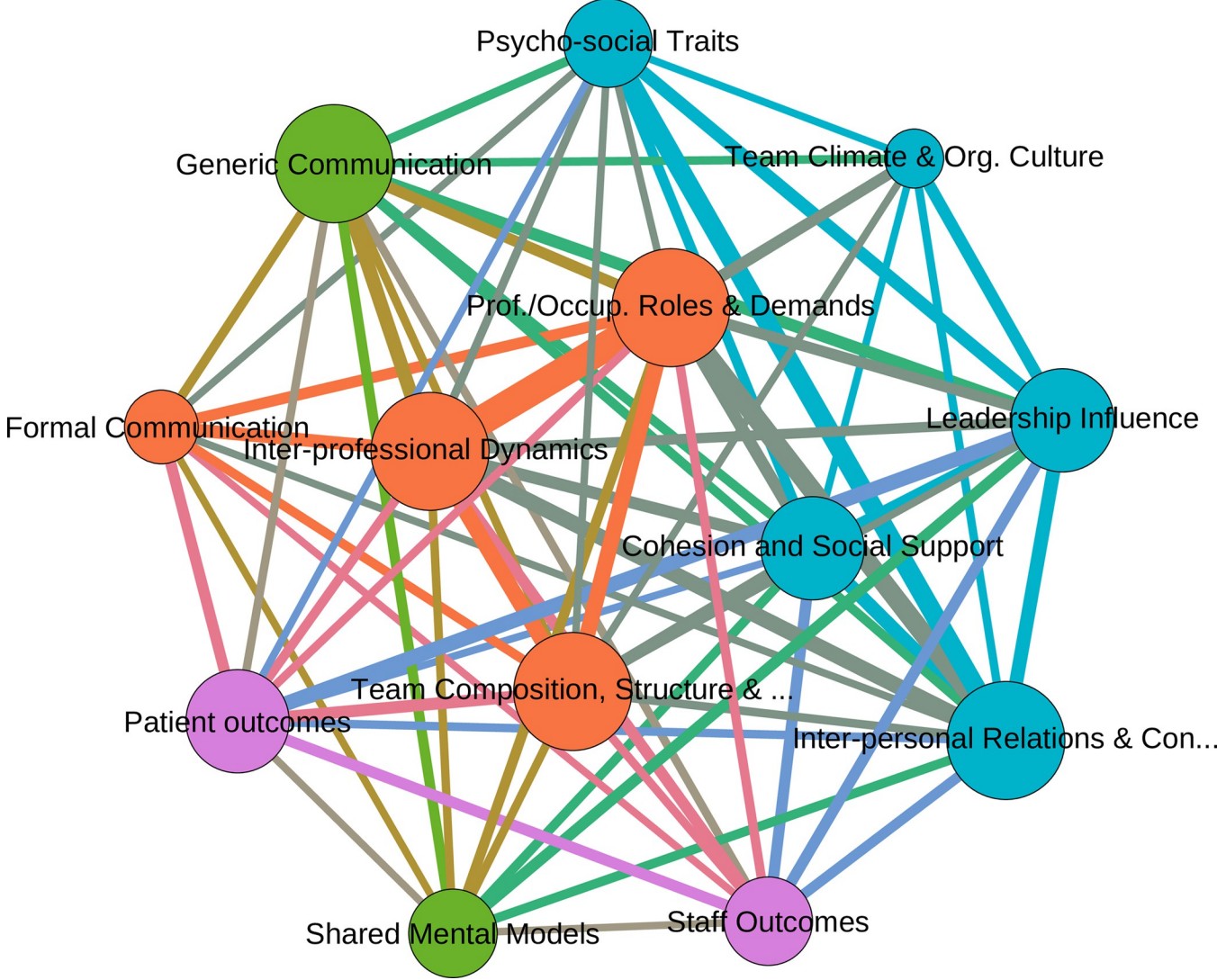

**Fig 4. Graphical representation of thematic relationships in reviewed papers.** The graph displays the relative importance of the themes by size of the nodes (based on the number of references), the relationships between nodes based on the physical closeness and the size of the connecting lines (derived from the number of references shared by individual themes) and the modularity of the themes (derived from the thematic closeness to each other). The graph further displays the presence of four thematic factors represented by different colours.

**Table 3. Overview of synthesised themes by factors.**

| Team- internal procedures and dynamics | Organisational & team-extrinsic influences |
|---|---|
| • Cohesion and Social Support <br> • Inter- personal Relations & Conflict <br> • Psycho-Social Traits & Personality <br> • Leadership Influence <br> • Team Climate & Org. Culture | • Formal Communication (e.g., huddles, meetings, checklists, whiteboards) <br> • Inter- professional Dynamics (e.g., hierarchy, power, prof. rivalry) <br> • Professional and Occupational Roles & Demands <br> Team Composition, Structure & Proximity |
| **Communicative processes** | **Team Outcomes** |
| • Generic Communication (e.g., information sharing, advice, seek help) <br> • Shared Mental Models | • Staff Outcomes (e.g., performance, satisfaction, retention, well-being) <br> Patient outcomes (e.g., mortality rate, length of stay, quality of care) |

individual factors (e.g., emotional intelligence, self-reflectiveness, confidence, communication style), attitudinal factors mediated by the team (e.g., accountability, commitment, values or enthusiasm), and lastly socio-economic factors (e.g., education, culture) were all found to influence individual's attitudes and behaviours vis-à-vis colleagues, impacting the work environment in which teamwork occurred [49–52]. This may suggest that an individual's personality and psycho-social traits impact teamwork mainly indirectly by influencing the work environment or team climate. Interestingly, the analysis implied that psycho-social traits and personality are most strongly linked to inter-personal conflicts and leadership (which is also evident in the network visualisation) suggesting that the personality and psycho-social abilities of leaders and colleagues impact upon the 'social climate' in inter-professional teams.

**Inter-personal relations and conflict.** Nine of the included reviews addressed the impact of inter-personal conflict and relationships on inter-professional or inter-disciplinary teams, with mixed results [18, 47, 49, 50, 52–55]. While some of the reviewed studies consider the importance of inter-personal relationships such as friendships between co-workers [48, 50, 51], few articles evaluated such factors systematically, with most of the emphasis on the presence of conflict or incivilities within the healthcare team environment. The reviewed evidence suggests that incivilities and disagreements are more likely between personnel from similar professional backgrounds with some medical specialties or professional groups being more prone to incivilities (e.g., nurses were reported as the most likely source of incivilities with other nurses) [51]. Similarly, being younger and less experienced female personnel was associated with a higher likelihood of being the recipient of incivilities [51]. Interactions between colleagues from different occupational backgrounds were predominantly dictated by the existence of inter-professional hierarchies or power dynamics and influenced by issues around the establishment of communication patterns, leadership, or cohesion [18, 48, 50, 51]. This suggests that the impact of inter-personal factors on teamwork is difficult to discern from the influence of inter-professional dynamics and pre-existing professional roles and demands [48, 50, 54].

**Cohesion & social support.** Ten articles referred to a socially supportive and cohesive team as instrumental in countering conflict within work teams [18, 47–51, 54–57]. However, few of them offered detailed evidence about the role of cohesion and social support on inter-professional teamwork. Results on the role of cohesion and social support for teamwork in inter-professional teams were mixed, with many studies addressing cohesion in the context of established teams only and as related to team building interventions. For example, results stemming predominantly from mono-disciplinary teams argued that continuous ward presence, working together for longer durations as well as developing a commitment to a unit were found to be associated with cohesion [48, 54, 56]. Furthermore, evidence indicates that spatial-temporal separation between team-members–as often the case between therapists, nurses, or doctors (e.g., separate break rooms, different work hours)–can lead to a decrease in cohesion in such teams [47, 54]. Nevertheless, conflicting evidence suggested that professional independence– while potentially hindering inter-professional collaboration–can nevertheless enable cohesive teamwork under pressure in inter-professional teams unfamiliar with each other [49]. The reviewed literature indicates that social support appears to be stronger when rendered by colleagues, rather than leaders [48]. This suggests that the development of a cohesive team-climate within the team is instrumental not only to increase teamwork and performance, but also to provide an environment in which team-members can aid each other in communicating problems, manage needs and aid in coping with stress [18, 48, 50].

**Leadership influence.** Nine of the reviewed studies observed the impact of leadership on healthcare teams [16, 18, 47, 49–51, 55–57]. The papers list a range of leadership behaviours such as injustices, not listening to input of subordinates and reporting to multiple managers or

leaders that have been associated with poor communication, lack of trust and social support [47, 49, 50]. A range of different behaviours were linked to effective leadership, such as using inclusive language, appearing calm, sharing knowledge, flattening hierarchies, allowing members to ask for advice and involving them in decisions, or creating a joint understanding [18, 47]. Finally, leadership was described in several papers as having a positive effect on the creation of shared mental models, psychological safety among team members, and reduction of incivility and team conflict in acute healthcare teams [18, 50, 51]. However, few of the reviewed papers provided a differentiated view of leadership in mono-professional, inter-disciplinary or inter-professional clinical teams. The little evidence provided suggests that leadership across inter-professional boundaries has been found to be subject to a range of problems associated with occupational demands and job expectations, which may increase the risk for leadership to 'slip into hierarchies' [47] because of professional differences between doctors, nurses and therapists. In such teams, several leadership styles have been found to negatively impact teamwork and communication, such as dominant, passive, transactional or authoritative leadership styles, while more collaborative and encouraging styles such as shared leadership (SL) or transformational leadership were found to be more beneficial for inter-disciplinary teamwork and team cohesion [18, 49, 50]. Intra-disciplinary leadership was linked to increased information exchange and clear shared mental models, while inter-professional leadership and shared leadership was associated with higher team performance and more positive team climate [18, 49, 51].

**Team climate and organisational culture.** Ten of the 18 reviewed articles discussed organizational culture or support structures, involving the impact of organisational culture on team climates, professional educational development, leadership or team training interventions [47–51, 54–58]. While several studies mentioned the importance of organisational or institutional factors–influenced by healthcare and hospital systems and regulations–for effective teamwork and collaboration in interprofessional or interdisciplinary teams, only a few explicitly integrated organisational aspects into their synthesis [47, 51, 57]. Most of the reviews instead discussed the impact of organizationally mandated teamwork interventions targeting teamwork processes in acute settings (e.g., simulation or leadership training interventions) [16, 53, 57, 59]. Nevertheless, aspects of organisational culture were identified to explain key characteristics of team processes and their impact on effective teamwork [48, 51, 52, 60]. It was concluded that supportive organizational culture, which encourages teamwork and diversity are associated with decreased levels of incivilities [48, 49, 51] and may impact team effectiveness by influencing team processes, norms and task design [57]. For example, individuals' willingness to participate in dialogue pertaining to patient safety (e.g. daily safety huddles) was determined by the wider safety culture and the extent to which individuals felt safe to freely exchange information and discuss issues surrounding patient safety [52]. Furthermore, educational interventions provided by the organization and a commitment to training and professional development were argued to enhance communication while organizational cultures and hierarchies within segmented departments may impact the utilization of inter-professional teams due to lack of role understanding and education [47, 50, 54, 58].

## Factor 2: Communicative processes

The second factor identified in the graphical network visualisation, pertained to communication and the establishment of shared mental models within the team. Repeated modular computation confirmed that formal communication processes–while addressing communication–were not included in this cluster, suggesting that such formalised processes (identified through particular processes (e.g., bedside rounds, meetings) as well as written procedures (e.g., health records, whiteboards) are different than informal communicative procedures in teams.

**Generic communication.** Key to many of the above discussed factors was the role of communication on inter-professional and inter-disciplinary teams in intensive and acute care environments [16, 18, 47–50, 53, 54, 60]. While the papers appeared to show widespread agreement over the importance of communication, they differed with regards to levels of analysis and the function of communication, painting a diverse picture of the role of communication for patient outcomes, establishment of commonality and clarification of roles and responsibilities. In inter-professional teams where team-members do not know each other prior to allocation to the team, coordination of activities appears to be particularly crucial for teamwork [49, 54]. As such, inter-professional teams may suffer from inherent dominance of individual professional groups (such as medical doctors) leading to decreased communication across professional or hierarchical boundaries [47, 49, 52, 55]. Addressing such communication barriers in inter-professional teams, [54] suggested a model for successful communication and collaboration is required, arguing that effective communication relies upon a shared realisation by team-members of the need and benefits of information sharing, the capacity of team-members (e.g., time, shared understanding) and the opportunity for information sharing (e.g., space, documentation).

**Shared mental models.** Eight of the reviewed papers discussed communication in terms of cognitive effects on interprofessional teams, by establishing a team-wide understanding and knowledge base of roles, terminology, task facilitation and attainment [18, 47, 49, 53, 54, 59, 60]. The formation of these "shared mental models" was described as a requirement for effective teamwork and successful team performance in healthcare teams, allowing increased recognition of shared problems, familiarity with team-members' roles and responsibilities, and anticipation of team-member's needs [49, 53, 54, 59]. Whilst most of the evidence on shared mental models was based on mono-professional healthcare teams, a few of the reviewed studies highlighted important aspects for the development of shared mental models within inter-professional or interdisciplinary teams. For example, Aufegger et al., described inter-professional teams benefiting from frequent information exchange, shared responsibility patterns as well as process and equipment coordination to improve information processing, planning and decision making between team-members [18]. Another review highlighted the utility of shared inter-professional vision statements as facilitating shared identity and team values [47]. Indeed, in teams with fluctuating or infrequent attendance to formal communication sharing activities (e.g., meetings), shared mental models were found to be essential to effective teamwork [48, 54].

## Factor 3: External team occupational and organisational influences

The thematic analysis, supplemented by the network analysis identified four occupational or organizational themes, which are predominantly influenced by professional or occupational demands and role responsibilities, or by organizational and institutional decisions about the structure and processes within the team.

**Inter-professional dynamics.** Thirteen of the 18 reviewed studies discussed inter-professional dynamics, pertaining to the existence of power dynamics, formal or informal hierarchies, and professional rivalries as a key issue in inter-professional teams [18, 47–51, 53–56, 58–60]. Health professionals' experience and views on interprofessional collaboration appear to reflect their contribution in clinical decision making and development of ward processes [48, 50], with the distribution of power within healthcare teams being strongly associated with individuals' professional experience, professional roles, and power relations within teams. The significance of power relations for team collaboration–which in many cases is influenced by legal and professional regulations–is widely discussed within the observed literature examining

power imbalances within healthcare teams and its impact on teamwork and team climate [47, 48, 50, 54]. Evidence shows that physicians' perceived superiority (due to longer educational pathways) and the perceptions of emotional aspects of care as less clinically relevant (tasks that are prevalent within nursing teams) can undermine inter-professional teamwork [47, 52, 60]. Similarly, team-members perceived to be lower in the hierarchy (e.g., junior members, or nurses vis-à-vis doctors) can be inhibited in speaking up during meetings [52]. Such assumptions can place nurses, allied health professionals, and social workers in a much less powerful position within their teams with reduced accountability and limited input in clinical decision-making [47, 48, 50]. Importantly, interprofessional power dynamics can fracture team cohesiveness, which is dependent on the quality of interpersonal relationships within the team, and the level of trust that exists among team members [18, 48, 50, 51].

**Professional and occupational roles and demands.**   Given the importance of different occupational backgrounds for inter-disciplinary and inter-professional teams many included reviews discuss how team members' professional roles, responsibilities and demands can impact on the quality of interprofessional team collaboration [18, 47–52, 54–56, 58, 60–62]. For example, distinctive role characteristics (e.g., nurses' continuous ward presence, medication rounds, or patient care requirements) as well as time constraints, can severely limit physical participation in meetings or consistent participation in inter-disciplinary working arrangements [48, 54, 60]. Similarly, therapists, who–unlike nurses–may have more clearly outlined professional duties and demands (e.g., patient rehabilitation) can be limited by differing working hours, transient group membership and physical separation, which can decrease successful team-collaboration and limit understanding of roles and responsibilities [47, 54]. The thematic analysis therefore identified that role ambiguity and unequal expectations, lack of shared goals, the transactional nature of inter-disciplinary collaboration, as well as associated accountability struggles were often cited as a source of conflict leading to poor unit morale, fractured communication, and limited identification with one's team [49, 50, 60]. Similarly, occupational characteristics, such as workload and staffing, were frequently reported as impacting upon inter-professional teamwork [50]. Such dynamics may cause team-members to under-utilise inter-professional colleagues in favour of their own professional group [49, 58] and limit familiarisation with team-members, which may impact the development of relationships with other professionals and contribute formally or informally to decision-making [54, 56].

**Team composition, structure & proximity.**   Team composition and spatial-temporal characteristics of team structures were discussed in 12 of the reviewed articles, showing that optimal team integration and collaborative work can differ across different settings and types of teams [16, 47–50, 54–56, 58–61]. For example, collaboration within mono-professional teams can be less challenging than teamwork within inter-professional and inter-disciplinary teams whose members come from different professional backgrounds [56] and may have different processes and organisational hierarchies. In this environment, team stability as well as proximity of team-members was found to be central to the understanding of inter-professional team processes and how they may impact teamwork [49, 54, 55, 60]. For example, physical space and constraints can determine the quality of ties that are formed within teams [54], while physical separation, asynchronous work schedules, and associated limited interaction can impede team cohesion and sharing of information [47, 56]. Unfortunately, most of the reviewed studies focused on teams with stable membership, with only one study addressing members of acute teams, who were described as operating more autonomously than team members in non-acute settings due to their infrequent close collaboration outside of "periods of crisis" [47]. The characterisation of these acute teams as forming identifiable social units able to engage in highly collaborative work during periods of crisis means that they can exhibit

efficient and swift responses to urgent medical situations in a complex interdisciplinary environment [16, 18, 59]. Similarly, the effectiveness of emergency response teams is dependent on hospitals' staffing decisions, appropriate team composition decisions (e.g. an outreach nurse operating in liaison with critical care), systems' frequent utilisation and team members' continuing education on and adjustment to the required processes [58, 62].

**Formal communication processes.** Several studies addressed formal communication processes developed within interprofessional teams. These included a range of communication processes with varying formats, length and modality, e.g., interprofessional ward rounds, huddles, case conferences, handovers, debriefing, informal bed side meetings, documentation, reports, checklists [50, 52, 54, 59, 60]. However, while there appears to be widespread evidence for the efficacy of such communication processes, various reviews highlighted how differing interprofessional roles and responsibilities limit such effective communication. For example, distinctive organizational demands, differing schedules, lack of time, and dual responsibilities (e.g., physical care or medication rounds) restrict the ability of personnel to attend formal and informal meetings [52, 54], thus diminishing the positive effect of such processes. Similarly, successful implementation of interprofessional working practices such as interdisciplinary bed rounds require a culture that encourages and appreciates input from every multidisciplinary team member [60]. For example, Buljac-Samardzic et al. (2020) suggest that simple tools such as checklists, goal sheets and case analysis can improve communication, information exchange, accountability, and patient care.

## Factor 4: Team outcomes

The last factor identified through the network visualisation pertained to themes around staff and patient outcomes.

**Patient outcomes.** Most of the studies assessing patient outcomes focused on the impact of specific processes on patient outcomes e.g., interdisciplinary bedside rounds, huddles, teamwork or leadership training, with few allowing an insight into the effect of inter-disciplinary or inter-professional team composition on patient outcomes [16, 18, 47, 48, 51–54, 56, 59–61]. While some preliminary findings supported the assertion that team and leadership training may have some beneficial impact on patient outcomes, the observed reviews showed mixed results with regards to improving patient outcomes in acute healthcare environments [16, 53]. Nevertheless, some findings indicated that multi-professional teams may be highly influential in affecting patient outcomes and driving quality and safety cultures in healthcare settings [18, 47, 61]. Correspondingly, spatial-temporal design, organizational hierarchies and management were reported to impact upon patient care and teamwork [47, 49, 56]. For example, the inclusion of pharmacists in inter-professional intensive care teams has shown improved ICU care and reductions in mortality, length of stay and adverse drug events [61]. Similarly, inter-, or multi-disciplinary team structures decreased hospital admissions and readmissions, increased patient satisfaction levels and integration of patient services and, in surgical environments, may improve hospital survival and reduce cardiac arrest rates [48, 62].

**Staff outcomes.** Collegiality, commitment to one's role and role clarity, communication within the team, empowerment and relational co-ordination are associated with positive staff outcomes, such as teamwork experiences, desirable workplace outcomes and positive indicators of occupational wellbeing [18, 48, 55, 60]. For example, effective communication in healthcare teams is related to patient outcomes, such as improved coordination and facilitation of care, patient satisfaction, safety culture, and shared decision making, as well as decreased hospitalisation costs, length of stay and medical failures, and readmission rates [52, 54, 55, 59, 60]. Implicit outcomes of communication were linked to enhancing team identity, improving

trust, reducing conflict and increasing teamwork and team effectiveness [48, 50]. Similarly, social support was reported as allowing team-members to work more collaboratively, cultivate a shared sense of responsibilities and understanding of goals and tasks, increase psychological safety and create a positive working environment [18, 50]. Social support from colleagues and supervisors within nursing teams was reported to improve job satisfaction [48], while colleagues' support in the form of mutual concern for colleagues' well-being and resilience (ahead of the organisation and supervisors), was found to be highly important when recovering from traumatic experiences [18, 48]. Conflict reduction was associated with increasing staff satisfaction and performance [48, 55]. Importantly for this analysis, staff turnover was found to correlate with intra-group conflict [48], suggesting that leaders need to be aware of such risks. Schmutz et al. (2019) highlighted that teams that engage in teamwork processes are 2.8 times more likely to achieve high performance than those that do not, while healthcare teams of all sizes and levels of acuity of care can benefit from teamwork processes. Furthermore, understanding the characteristics of effective teamwork processes is important as it is the shared activities of interprofessional and multi-professional teams that can count for patient safety and the occurrence of medical errors [49]. Overall, team composition and especially team size, familiarity with procedures, and mix of technical competencies can play a vital role in team effectiveness [49]. As such, relational and structural teamwork characteristics (i.e. interpersonal interactions, power dynamics, team proximity and job role characteristics) appear to shape the quality of teamwork and impact significantly on team communication, team cohesiveness, effectiveness, and staff wellbeing [18, 47, 48, 51, 54, 55, 60]. Associations between clinicians' positive indicators of teamwork and positive and negative indicators of occupational wellbeing (e.g., work engagement and burnout) showed that those that perceived higher quality of teamwork also reported higher occupational well-being or less strain [55].

## Discussion

The thematic analysis of the selected systematic reviews identified 13 themes related to teamwork in rapidly deployed interprofessional teams in intensive and acute care settings, which could be organized into four factors. The first of these four factors, addressing team-internal factors, consisted of themes around the impact of psycho-social traits on the work environment and team climate, the role of interpersonal relations and conflict, team cohesion and social support as well as leadership. The literature suggests that while personality impacts teamwork indirectly, by influencing an individual's attitude and behaviour vis-à-vis colleagues, the thematic analysis also implied that psycho-social traits and personality are most strongly linked to inter-personal conflicts and leadership suggesting that the personality and psycho-social abilities of leaders and colleagues' impact upon the climate in inter-professional teams. Simultaneously, conflict in inter-professional or inter-disciplinary teams tends to lead to fewer open incivilities or rude behaviour than seen in teams from similar professions [49–52], implying that conflict in such teams is likely perceived through the lens of occupational identities, medical hierarchies, and power dynamics based on occupational sub-divisions. Lastly, teamwork and cohesion reportedly are increased through continuing ward presence and co-habitation of office space [47, 54], while social support, rendered by immediate colleagues and superiors, provides an environment in which team-members can aid each other in communicating problems, manage needs and aid in coping with stress [18, 48, 50].

The second factor pertained to communicative processes, showing that lack of opportunities for formal and informal information exchange can hinder the coordination of activities and the development of shared responsibility across inter-professional teams and limits the development of shared mental models, which is essential for effective teamwork. As such,

team-wide communication during handovers or even during breaks may be crucial for team-members to develop professional and personal familiarity, recognise shared problems, and anticipate team-member's needs. Effective communication also appears to impact upon enhanced coordination and facilitation of care, patient satisfaction, reduced medical errors and lower readmission rates, while a wider integration of skill-mix can reduce mortality, improve survival and cardiac arrest rates, shorten hospital length of stay and reduce adverse drug events [18, 50].

The third factor comprised of team-external factors, which stem from different professional responsibilities and occupational demands, as well as organizational structures and processes related to health and safety protocols, workforce allocation models, or departmental layout, which impact upon the functioning of the team [18, 47–52, 54–56, 58, 60–62]. Differences between physicians, therapists and nurses' perception of patient care and their involvement in clinical decision-making may cause grievances and conflict in inter-professional teams [18]. Implicit stereotypes about occupational role expectations, by which health care personnel are perceived as being solely responsible for specific aspects of care (e.g., emotional, and patient support, family liaison, or rehabilitation) hinder the development of shared identity and cohesion and thus undermine inter-professional teamwork. Simultaneously, ambiguity and unequal expectations, lack of shared goals, transactional inter-disciplinary collaboration, or mismatch of accountability appear to influence poor unit morale, fractured communication and team identification [49, 50, 60]. It is therefore proposed that for rapidly deployed interprofessional teams in intensive and acute care inter-professional communicative arrangements, such as use of checklists or scripts, health records, or interdisciplinary bedside rounds, as well as informal information exchange can improve role clarity and collaboration [54, 60]. Simultaneously, frequent staff turnover, departmental separation, or asynchronous work schedules can limit interactions between team-members and encumber team cohesion and opportunities for information sharing [47, 54].

These findings provide a framework by which important lessons from the experience of inter-professional teams in a COVID environment can be analysed and interpreted. This is likely to be even more complex, where inter-professional team-members may struggle to develop cohesive ties due to their short duration within the teams and such cohesion may be disrupted, both in the team(s) personnel are deployed to and the team(s) personnel are deployed from. Given that much intra-team conflict between members from different occupational or professional backgrounds is perceived through the lens of pre-existing occupational identities, managers and leaders of such teams need to counteract ostracization and team separation based on occupational membership. The reviewed literature suggests inherent power-dynamics can be further mitigated by clarifying roles and responsibilities of team-members and emphasizing the individual contribution and skills of each member in the team. Managers and leaders in crisis situations may do well to utilise and emphasize the professional skills and abilities of their inter-professional team-members to increase cohesion in crisis situations such as COVID-19 deployment. Similarly, the literature proposes that inter-professional teams may be able to develop shared mental models through more shared meetings, enhanced information exchange (utilising both verbal and technological communication), while clearly identified formalised communication procedures (e.g., whiteboards, checklists, health records) may improve role clarity, collaboration, information exchange, accountability, and patient care [54, 60]. Given the issues around the spatio-temporal location of fluid team-members, the review suggests that organizational decisions about workforce allocation, physical space, team composition, team size, and skill mix of inter-professional team-members can impact the teamwork and team cohesion in inter-professional COVID wards.

## Limitations of this study

Most of the literature reviewed was based on empirical studies in developed economies–only six of the 18 included reviews including any individual studies from the Global South—limiting the generalisability of the evidence. Another limitation of the study pertained to terminological and ontological differences, which made the selection of studies difficult. With many studies reviewing evidence from empirical studies including mono-professional, multi-disciplinary as well as inter-professional teams, comparison, synthesis of evidence from the literature is somewhat hampered. Furthermore, many of the studies were of differing quality and most of the studies reviewed focused on either permanent or sporadic teams amalgamated for the purpose of training exercises and simulation, thus reducing the generalisability of some of the findings for ad-hoc, fluid inter-professional teams, such as those deployed during COVID-19. A final limitation pertains to the utility of a graphical network visualisation in systematic reviews. Due to the research design deliberately including different methodological approaches (e.g., quasi-experimental studies, observational studies, longitudinal studies, mixed methods studies, randomized controlled trials, and qualitative studies) our analysis was restricted to a thematic and visual analysis of the data. To mitigate this restriction, we utilised a combined approach, combining a thematic analysis with a graphical network visualisation, to extrapolate meaningful factors in the data. Nevertheless, it is important to note that the visualisation cannot be taken as an exact reflection of the importance of a particular node for the general discussion on teamwork in healthcare teams. Rather, it offers supplementation of the thematic analysis and similarly as a subjective product of the selection criteria and the thematic coding. For example, the presentation of themes in the graphical network analysis (e.g., the size and distance of shared mental models or proximity) may therefore be more emblematic of a general dearth of studies addressing their significance in inter-professional or inter-disciplinary health-care teams.

## Conclusion

This systematic review has provided an overview of different challenges and difficulties for effective teamwork in inter-professional healthcare teams that have to be rapidly formed for work in an acute setting. The review provides an important overview of 18 different reviews, that resulted in four different factors and 13 themes derived from the literature. This can guide the further analysis of inter-professional teams in emergency and crisis deployment. Many of these themes have been found to impact on both staff and patient outcomes, with lack of inter-professional and inter-disciplinary integration and teamwork not only decreasing patient care and patient safety, but also to increase stress and conflict in inter-professional team-members. Collegiality, role commitment, role clarity, communication within the team, empowerment and relational co-ordination are related to positive well-being and more optimistic team climate, while increased interpersonal interactions and communication and team proximity influence the quality of teamwork, improve cohesion, and trust, increase team effectiveness and a shared team identity. The choice of applying an integrative narrative synthesis process [40], utilising the thematic coding supplemented by graphical network analysis proved highly beneficial, as it not only allowed to make the selection process of themes more transparent, but permitted the identification of underlying factors in the literature on inter-professional teams, as well as enable the visual representation of themes in the literature. The utilisation of such an approach is recommended for future systematic reviews.

These findings imply that managers and team leaders in fluid and ad-hoc created inter-professional healthcare teams in an intensive care environment need to pay attention to minimising pre-existing occupational identities and power-dynamics by emphasizing skill mix,

establishing combined workspaces and break areas, clarify roles and responsibilities, facilitate formal information exchange and developing informal opportunities for communication. Nevertheless, due to the lack of evidence on ad-hoc, fluid teams in crisis situations–such as COVID-19 –it is clear there is a need for more research to empirically examine the factors proposed in this article and to address in detail how rapidly formed teams develop effective teamwork and to understand how they overcome barriers to successful patient care under extraordinarily stressful circumstances. This research would not just help healthcare systems deal with the current pandemic but contribute to the successful formation of rapidly formed teams in future crisis situations.

## Supporting information

**S1 Checklist. PRISMA 2020 checklist.**
(DOCX)

**S1 File. List of articles used in study.** This File contains a complete list of the selected articles used in the analysis.
(DOCX)

**S2 File. Visualization data.** This spreadsheet contains three tabs: (1) the matrix of references as downloaded from NVivo, (2) the Nodetable and (3) the Edgetable, necessary for graph model development in Gephi. With these tables the graph (Fig 4) can be reconstructed.
(XLSX)

**S1 Table. Overview of inclusion and exclusion criteria.**
(DOCX)

**S2 Table. Contribution of articles to themes.**
(DOCX)

**S3 Table. Types of healthcare teams, occupation of team members, and clinical settings.**
(DOCX)

## Author Contributions

**Conceptualization:** Stefan Schilling, Maria Armaou, Zoe Morrison, Paul Carding, Martin Bricknell, Vincent Connelly.

**Data curation:** Stefan Schilling.

**Formal analysis:** Stefan Schilling, Maria Armaou.

**Funding acquisition:** Stefan Schilling, Vincent Connelly.

**Investigation:** Stefan Schilling, Maria Armaou, Zoe Morrison, Paul Carding, Martin Bricknell, Vincent Connelly.

**Methodology:** Stefan Schilling, Maria Armaou.

**Project administration:** Stefan Schilling.

**Software:** Stefan Schilling.

**Supervision:** Stefan Schilling, Vincent Connelly.

**Validation:** Stefan Schilling, Maria Armaou, Zoe Morrison, Paul Carding, Martin Bricknell, Vincent Connelly.

**Visualization:** Stefan Schilling.

**Writing – original draft:** Stefan Schilling, Maria Armaou.

**Writing – review & editing:** Stefan Schilling, Maria Armaou, Zoe Morrison, Paul Carding, Martin Bricknell, Vincent Connelly.

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
