## [Decision Letter · Decision Letter 0]

13 May 2022

PONE-D-22-01696Understanding teamwork in rapidly deployed inter-professional teams in intensive and acute care: a systematic review of reviewsPLOS ONE

Dear Dr. Schilling,

Thank you for submitting your manuscript to PLOS ONE. After careful consideration, we feel that it has merit but does not fully meet PLOS ONE’s publication criteria as it currently stands. Therefore, we invite you to submit a revised version of the manuscript that addresses the points raised during the review process.

Together with both reviewers, I congratulate the authors for this nicely developed piece of research. I would recommend, however, shortening some sentences as suggested by one of the reviewers and to facilitate the reading of the final version of the text. 

We look forward to receiving your revised manuscript.

Kind regards,

Sara Rubinelli

Academic Editor

PLOS ONE

Journal Requirements:

2. We note that you have referenced (ie. Bewick et al. [5]) which has currently not yet been accepted for publication. Please remove this from your References and amend this to state in the body of your manuscript: (ie “Bewick et al. [Unpublished]”) as detailed online in our guide for authors

Reviewers' comments:

Reviewer's Responses to Questions

**Comments to the Author**

1. Is the manuscript technically sound, and do the data support the conclusions?

Reviewer #1: Yes

Reviewer #2: Yes

2. Has the statistical analysis been performed appropriately and rigorously? 

Reviewer #1: I Don't Know

Reviewer #2: N/A

3. Have the authors made all data underlying the findings in their manuscript fully available?

Reviewer #1: Yes

Reviewer #2: Yes

4. Is the manuscript presented in an intelligible fashion and written in standard English?

Reviewer #1: Yes

Reviewer #2: Yes

5. Review Comments to the Author

Reviewer #1: The authors provide a comprehensive summary of the available evidence and should therefore be applauded. The graphical display of node-networks is helpful and should be integrated into the main manuscript. The authors tend to use very long sentences. I would recommend to review the manuscript for the possibility to shorten sentences or make it several sentences (e.g. first paragraphs of the discussion).

Reviewer #2: This is a well-written and well-researched systematic review of reviews. The process for selecting and carrying out the reviews was transparent, clear, and thoroughly described. The themes found within the final 18 articles, relating to the four overarching factors, were clearly explained and the relationships between them well-established. Ultimately, the conclusions drawn were were logically derived from the factors/themes discovered. To me, this was an exceptionally well done review.

6. PLOS authors have the option to publish the peer review history of their article (what does this mean?). If published, this will include your full peer review and any attached files.

Reviewer #1: **Yes: **Thilo von Groote, MD

Reviewer #2: No

---

## [Author Response · Author response to Decision Letter 0]

14 Jul 2022

Dear Thilo and Reviewer #2: 

Thank you very much for reviewing our article and for your kind, courteous and constructive feedback. Your insights were much appreciated.

We have addressed your points in the revised manuscript, with a focus to making the sentences shorter. 

With best Wishes and many thanks, 

Stefan

---

## [Editor Report · Decision Letter 1]

1 Aug 2022

Understanding teamwork in rapidly deployed inter-professional teams in intensive and acute care: a systematic review of reviews

PONE-D-22-01696R1

Dear Dr. Schilling,

We’re pleased to inform you that your manuscript has been judged scientifically suitable for publication and will be formally accepted for publication once it meets all outstanding technical requirements.

Kind regards,

Sara Rubinelli

Academic Editor

PLOS ONE
---

## [Editor Report · Acceptance letter]

9 Aug 2022

PONE-D-22-01696R1 

Understanding teamwork in rapidly deployed interprofessional teams in intensive and acute care: a systematic review of reviews 

Dear Dr. Schilling:

I'm pleased to inform you that your manuscript has been deemed suitable for publication in PLOS ONE. Congratulations! Your manuscript is now with our production department. 

Kind regards, 

on behalf of

Dr. Sara Rubinelli 

Academic Editor

PLOS ONE